# COVID-19: The Ethno-Geographic Perspective of Differential Immunity

**DOI:** 10.3390/vaccines11020319

**Published:** 2023-01-31

**Authors:** Usman Abdullah, Ned Saleh, Peter Shaw, Nasir Jalal

**Affiliations:** 1Department of Biomedical Sciences, Pak-Austria Fachhochschule, Mang, Haripur 22621, Pakistan; 2Synsal Inc., San Jose, CA 95138, USA; 3Oujiang Lab, Zhejiang Lab for Regenerative Medicine, Vision and Brain Health, Wenzhou 325000, China

**Keywords:** COVID-19, copy number variation (CNV), virome, microbiome, endoretroviral genome (ERV), geographic disparity

## Abstract

Coronavirus disease 2019 (COVID-19), the agent behind the worst global pandemic of the 21st century (COVID-19), is primarily a respiratory-disease-causing virus called SARS-CoV-2 that is responsible for millions of new cases (incidence) and deaths (mortalities) worldwide. Many factors have played a role in the differential morbidity and mortality experienced by nations and ethnicities against SARS-CoV-2, such as the quality of primary medical health facilities or enabling economies. At the same time, the most important variable, i.e., the subsequent ability of individuals to be immunologically sensitive or resistant to the infection, has not been properly discussed before. Despite having excellent medical facilities, an astounding issue arose when some developed countries experienced higher morbidity and mortality compared with their relatively underdeveloped counterparts. Hence, this investigative review attempts to analyze the issue from an angle of previously undiscussed genetic, epigenetic, and molecular immune resistance mechanisms in correlation with the pathophysiology of SARS-CoV-2 and varied ethnicity-based immunological responses against it. The biological factors discussed here include the overall landscape of human microbiota, endogenous retroviral genes spliced into the human genome, and copy number variation, and how they could modulate the innate and adaptive immune systems that put a certain ethnic genetic architecture at a higher risk of SARS-CoV-2 infection than others. Considering an array of these factors in their entirety may help explain the geographic disparity of disease incidence, severity, and subsequent mortality associated with the disease while at the same time encouraging scientists to design new experimental approaches to investigation.

## 1. Introduction

The global pandemic, nicknamed coronavirus disease 2019 (COVID-19), is a disease that is caused by the severe acute respiratory syndrome coronavirus 2 (SARS-CoV-2). The virus has caused about 84 million confirmed cases of COVID-19 and 880,000 confirmed deaths until the filing of this article [1]. Interestingly, 64% of the morbidities (i.e., 55 million) could be linked to race/ethnicity, while 85% of the mortalities (i.e., 750,000) could be linked to race/ethnicity [2]. The world has been fighting this pandemic by developing new treatment methods while also trying to administer vaccines targeted at achieving herd immunity in public. About 10,925,055,390 doses of various vaccines (RNA, protein, and attenuated-virus-based) have been manufactured and shipped worldwide, causing the pandemic to slow down, but it is far from being over [1]. SARS-CoV-2 is a betacoronavirus with a positive sense, single-stranded RNA about 30 Kb in size [2]. The virus causes infection of the upper respiratory tract in mild infections, but as the disease progresses, pneumonia and hypoxemia may develop. A patient suffering from a severe version of the disease may develop acute respiratory distress syndrome (ARDS), shock, encephalopathy, myocardial injury, and acute kidney failure [3]. Some risk factors already identified for severe and potentially fatal COVID-19 illness include age, sex, and pre-existing medical conditions such as hypertension, diabetes mellitus, cancer, and autoimmune conditions. We have observed a disparity in the general severity and mortality rates of COVID-19 across various geographic regions of the world. Nations such as Bangladesh and Pakistan, which were previously predicted to be on the worst end of the pandemic, remain seemingly unimpaired, all the while exhibiting low numbers of both mortality and severity of the disease [4]. This is despite these developing countries having relatively poor healthcare infrastructures and high population densities, accompanied by poor hygienic conditions among the population, which essentially create a perfect breeding ground for communicable respiratory diseases such as COVID-19; thus, the hitherto predicted high viral transmissibility rates (R_0_) [4]. However, the actual morbidity, mortality, and transmissibility rates are noticeably low in these developing countries, whereas if we consider the western, more developed countries in Europe and North America, the severity of the disease and its mortality rates are comparatively much higher, with mortality rates reaching as high as 30 percent in some well-developed countries, such as Belgium, during different phases of the pandemic [5]. With the risk of viral transmission rising in these regions, more infectious strains/variants have appeared, such as omicron [6]. Some studies have even suggested that 37 percent of all mortalities of the pandemic came from the region known as the European Union (EU) [7]. In contrast, the EU only accounts for about 9.78% of the world’s population [8].

Despite these discouraging statistics, Europe has a much better healthcare infrastructure, hygienic conditions, and better health awareness than developing countries such as Pakistan and Bangladesh. Until now, the biggest argument used to justify this disparity was the population’s average age in these regions. In the case of Pakistan, about 60% of the population was recorded to be below the age of 30 years [9]. However, in this review, we argue that there must be some other biological factors at play that may also contribute to this disparity. It is important to mention that the following article is more about proposing a viable hypothesis based on previous studies and the authors’ deductions. However, all forthcoming hypotheses defend the demographic disparity idea; in other words, we indicate additional biological mechanisms that add to the variability of outcomes among populations from different demographic regions. We also briefly discuss the role of these biological phenomena in viral infectability and repercussions for the more comprehensive host immune system and present evidence from the existing literature on how they show some variation among members of different populations.

## 2. Human Microbiota

The human body acts as a host for approximately trillions of pathogens: bacteria, viral particles, viruses, and bacteriophages that help us maintain various internal mechanisms of our body for homeostasis [10]. These organisms that collectively form the microbiota of the human body are, as yet, the unrecognized agents that play a crucial role in the overall working of our body. Some of those processes are:Digestion and the ability to harvest nutrients [11].Modulation of the appetite signal of the body [11].The production of important vitamins for the body (including vitamin K) [12].The regulation of epithelial growth and development [13].The metabolization of various xenobiotics in the body [13].Modulation of the host immune system [14].Modulating the human endogenous retrovirus (HERV) expression plays a role in modulating future antiviral immunity [15].

The bacterial, viral, and bacteriophage components of the microbiota are found in almost every body organ system, with the largest reservoir being the gastrointestinal system [14]. There have been studies proving the presence of viral microbiota detected in the oral cavity [14]. Healthy human lungs and respiratory systems also host these viral microbiota in the air passages [16]. The microbiota play a role in the human immune system (Figure 1) through the following main methods. 

The bacteriophages in our body help us fight against bacterial infections by directly targeting the virus particle [17].The most notable is how it fights off other incoming viruses for competition in the region while also immune priming the body in preparation for infection by external pathogenic viruses [16].The microbiota heavily influence innate lymphoid cells’ (ILCs) development, maintenance, and function [18]. The ILCs, as well as possessing all the characteristics of an innate immune cell, also have the characteristics of adaptive T cells to release inflammatory agents, but they lack antigen-specific receptors [19]. Furthermore, ILCs are diverse as they have five subsets: type 1, 2, and 3 ILCs, lymphoid tissue inducer cells, and natural killer cells [20]. Several studies have shown that ILC subsets are interconvertible if they receive the appropriate cytokine stimuli, demonstrating these cells’ high degree of plasticity [21]. They also have particularly unique roles in immunity that normal adaptive immune cells or innate immune cells do not possess on their own [22].Viral-induced IFN-I production helps against viral infections [23], whereas dysregulated IFN-I production exhibits unfavorable outcomes for COVID-19 through ineffective antibody production against SARS-CoV-2 [24,25,26].

The antimicrobial peptides such as the mouse Reg-IIIγ and its human analog HIP/PAP (a subset of the calcium-dependent C-type lectin superfamily) bind their bacterial targets via peptidoglycan and represent a primitive form of innate immunity. The expressions of these antimicrobial proteins require signals from the microbiota [27].

### 2.1. Human Microbiota and COVID-19

More recent studies have shown a direct correlation between human microbiota and disease progression, infection, and immunity. One study co-housed different strains of mice to generate hybrid microbiota and then analyzed the causation of colitis, disease progression, and immunity through a generalized microbe phenotype triangulation method. The study also found that mice with hybrid microbiotas had a bacterial family called Lachnospiraceae that provided protection from disease. Moreover, they reported that when colitis-prone mice were administered with *Colstridium immunis,* the bacterial species helped evade a colitis-associated death [28]. In another study, whole-metagenome shotgun sequencing data were used to reconstruct metagenome-assembled genomes (MAGs) from a total of 514 nasopharyngeal and fecal samples of patients with COVID-19 and controls. The method was used to reconstruct microbial MAGs to obtain non-redundant MAGs (nrMAGs). The gut microbiome signatures, thus obtained, could not only accurately distinguish healthy from diseased COVID-19 cases, but they could also predict the disease progression. A set of nrMAGs was identified that had a putative causal role in the clinical manifestations of COVID-19 and reported functional pathways that could potentially interact with the SARS-CoV-2 infection. Thus, a better understanding of the human microbiota is a prerequisite to evaluating the ethno-geographic perspective of differential immunity associated with COVID-19 [29]. In another study, an inverse correlation between *Bacteroidetes* abundance and the disease severity of COVID-19 was presented. A study on the murine gut showed the downregulation of ACE-2 expression by bacterial species of *Bacteroidetes* and *Firmicutes* phyla [30,31]. Studies have also shown that patients with more severe COVID-19 showed delayed and/or defective forms of type 1 interferons (products of microbiota that help in antiviral immunity) that lead to lung damage and ineffective anti-SARS-CoV-2 antibodies’ production [24,25,26]. There is also a correlation between the gut microbiota and the expression and activation of the inflammasome proteins [32,33], which are cytosolic multiprotein complexes that play a role in cell death and inflammation through the activation of caspases and hence lead to the maturation of proinflammatory cytokines following the detection of microbial and endogenous stimuli [34]. The neutrophil extracellular traps released by neutrophils due to inflammasomes have been shown to cause impaired ventilation due to excessive mucous viscosity in COVID-19 patients [35].

Many large-scale studies have illustrated the variation in the metagenomic and meta-communal microbiota, especially the virome data among people of different ethnicities and geographic locations [36,37]. Indeed, other predisposition factors exist, such as diet and age, but the geographic factors presented the most significant virome and overall microbiome variability [37].

After establishing a connection between the microbiome’s immune functions and the microbiota’s geographic variability [38], we present the hypothesis that the geographically variable viral microbiome may play a role in the regulation of severity and the mortality rates of COVID-19. 

As per our hypothesis, people in some regions may have a microbiome and virome that consists of microbes, viral particles, and phages that offer a fiercer immunological response and competition against SARS-CoV-2 and pneumococcus (cause of pneumonia and resultant co-morbidity associated with COVID-19), thus lowering the chances of mortality and disease progression into a more severe form; giving them an edge of survival over people of other geographical locations and vice versa.

### 2.2. Hygiene Hypothesis of Human Microbiota and COVID-19

For decades, the improvement in medical healthcare has helped humans to evade parasitic infections through the rampant use of antibiotics. This has caused a decrease in the diversity of native human microbiota that leads to the hygiene hypothesis, which proposes that there has been a gradual decline in microbial diversity due to hygiene, antibiotics, and urban living [39]. It further builds on the fact that most COVID-19 high-risk groups consist of individuals with pre-existing conditions such as obesity and diabetes, which are also associated with microbiome abnormalities.

## 3. Human Endogenous Retroviruses (HERVs)

Previously accepted and understood parts of junk DNA or parts of the “genomic dark matter,” known as endogenous retroviral genomic components, account for more than eight percent of our whole genome [40]. These endogenous retroviruses used to be part of ancient exogenous pathogenic viruses [41] that infected the cells of our ancestors but remained in the lysogenic phases and eventually became a part of the vertical host DNA [42] over time [43]. It was also previously thought that this part of our genome plays no role in the body’s normal functioning, but this belief could not have been more wrong as we now know, owing to more extensive research studies (Figure 2). This part of the human genome has a fundamental role in antiviral immunity, among other functions [44]. A study has successfully established a direct correlation between HERV expression and COVID-19. In a transcriptomic comparison of the bronchoalveolar lavage fluid (BALF) of COVID-19 patients with healthy controls and peripheral blood monocytes (PBMCs) from COVID-19 patients with controls, it was clearly shown that HERVs were dysregulated in the BALF of COVID-19 patients compared to healthy controls but not in PBMCs. The team also identified upregulated expression of multiple HERV families in senescence-induced HBECs in comparison with the non-induced HBECs, which is a clear indication of the differences in disease severity among age groups [45,46,47]. As evidence keeps mounting, the current understanding of an association between human endogenous retroviruses (HERVs) and antiviral immunity can be understood through the following mechanisms:

### 3.1. Activation of Innate Immune System

The long-misunderstood “Junk DNA” plays an important role in many functions. One of those functions is providing an antiviral immune response. Some of this junk DNA is also comprised of spliced endoretroviral (ERV) genome particles. These pieces of ERV act as a genetic memory and trigger the innate immune system sensing cascades through HERV-synthesized polypeptides [48]. An initial understanding of the HERV-induced immune proteins [49] is shown in Figure 3.

### 3.2. Production of Long Non-Coding RNAs (lncRNAs)

Regulating the antiviral immune response is achieved by producing long non-coding RNAs (IncRNAs) [50]. An example of this would be the production of murine endogenous retrovirus (ERV)-derived lncRNA termed lnc-EPAV that plays a role by boosting the antiviral gene expression. Other than that, it also increases the production of antiviral cytokines such as IFN-β and IL-6. The ERV pathway of the lnc-EPAV, being a positive feedback loop, also increases the antiviral immune response. Hence the splicing factor proline, glutamine-rich) that plays a role in splicing RNA molecules related to immune system proteins (SFPQ) also interacts with multiple HERV-derived RNAs, suggesting a similar immune response in humans [51].

### 3.3. Modulation of the Immune System by ERV-Based Proteins

Modulating the immune activation is accomplished by the production of immune-cascade-activating proteins [52]. This happens by the deliberate triggering of plasma-membrane-associated toll-like receptors (TLRs) by the retroviral proteins, which cause the release of massive amounts of IL-1β, IL-6, TNF-α, and IL-12p40. The retroviral proteins also cause the release of nitric oxide, which plays an essential role in the innate immune system (Figure 3 relates to step 15) [52].

### 3.4. Receptor Interference

Blocking the virus entry receptors by their envelopes is performed through a phenomenon called receptor interference [53]. The viral gene products of the ERV genome, called ENVs, are essential components of viral membranes that mediate receptor interference by binding to the host cell receptors before the pathogenic particle can bind. The ERV-derived ENVs (viral gene protein products) might as well bind to the newly synthesized entry receptors, thus preventing their appearance and transport to the cell surface, essentially closing the gateway of the virion entry into the host cell. The ENV proteins conformed with this phenomenon of receptor interference of avian leukosis viruses that were observed to induce resistance to the exogenous viruses of the same subgroup [54].

Angiotensin-converting enzyme II (ACE2) is the primary receptor of entry for SARS-CoV-2 inside human host cells [55]. Studies have shown that the virus could not enter the ACE2 receptor null host cells [56]. The spike (S) protein of the virus binds with high affinity to the ACE2 receptor, thus modulating the effective entry of the virus into the host cell [57]. Host proteins such as the Transmembrane Serine Protease 2 (TMPRSS2) are known to facilitate viral entry into the cell by cleaving the S protein, thus allowing for effective viral fusion with the host cell [58]. Consequently, we propose that if some proteins facilitate the entry of the virus, then there must be some proteins that also act as a blockade or interference to the receptor-based viral entry of SARS-CoV-2. There might be an ERV-based protein that could hence explain the low susceptibility of people from a certain population to the virus. Suppose that this hypothesis could be tested in a lab. In that case, it may allow us to use the presence of that protein to produce drugs that could upregulate the genetic expression of the proteins responsible for receptor interference or exogenously inject recombinant ERV proteins that mediate ACE2 receptor interference.

### 3.5. Complementing in a Negative Manner

With regards to complementing the external virion in a dominantly negative manner with their proteins [59,60], an example would be a reduction in the viral infectivity and susceptibility because of the murine *FV-4* (a gene that has shown a reduced susceptibility to murine leukemia virus in mice [60]).

### 3.6. Interference in the Replicative Cycle

Creating interference in the replicative cycles of the incoming viral particles is a phenomenon that was successfully demonstrated in a lab mouse gene called *Fv1*. This gene encodes a restriction fragment that inhibits the entry of the murine leukemia virus and seems to have been inserted into the murine genome by an ancient provirus about 45 million years ago. These studies also suggest that retroviruses are adapted to living with their hosts through the vertebrate evolutionary process [61,62].

### 3.7. Increase in the Diversity and Plasticity of Immune Genes

Increasing the plasticity of the immunity genes by a recombination process with the silent ERV genes [63,64] is a homologous recombination that may result in the loss of genes or gene fragments in monogenic systems. Nevertheless, let us consider the multigenic families. The risk-to-reward ratio is different, and the recombination or insertion of the ERV sequences may allow the immune system to facilitate the viral pathogen better. One such example would be the major histocompatibility complex (MHC) locus present in primates. Such involvement and recombination of the ERV genetic segments has resulted in the increased diversity of alleles and thus increased the spectrum of viral peptides that can be presented to T cells. This increased diversity in the viral peptides due to the diversity of alleles causes a more robust immune system that is better able to fight against diverse groups of pathogens [64,65,66,67,68,69].

### 3.8. LTRs-Based Enhancement of Antiviral Genes

When regulating the antiviral gene expression with their promoters and enhancers [70,71], there are a lot of promoter, inhibitor, and enhancer sites in the long-term repeats (LTRs) of the ERV, which promotes the encoding of antiviral proteins such as INF-α, etc. Thus, an individual’s immune system can be boosted or enhanced by the function of the ERV and LTR promoter and enhancer sites.

All the mechanisms mentioned above of the retroviral genome contribute to our defense against pathogenic infections, especially those caused by viruses. Furthermore, this biological factor, such as the one discussed above, is also variable among the populations of different geographical locations [72]. This leads us to speculate that HERV may add to the disparity observed in the severity and mortality rates of COVID-19 among members of different demographic locations.

### 3.9. Genetic Memory

Accumulated HERVs result from evolution and natural adaptation, and it would not be wrong to say that this component of the DNA may act as a form of genetic memory for the body. Since HERV is based on the genetic material of old retroviruses, it provides the body with access to the DNA and the protein components of these ancient viruses, which may, in fact, be used by the body to enhance its immunity in case it has to face a new generation of the ancient viruses that are already a part of its massive HERV database, which we refer to as the “immune memory” [73]. Therefore, the HERV acts as an important memory base for both the body and scientists who are trying to study ancient viruses [74,75,76].

### 3.10. Innate Immune System Triggering

The cells of the innate immune system become activated after detecting pathogen-associated molecular patterns (PAMPs). Two families, namely endosomal pattern recognition receptors (PRRs) and cytosolic PRRs are specialized in the recognition and detection of nucleic acids [77]. Detecting these nucleic acids quickly is an efficient way of fighting viral infections. ERVs trigger the same sensing of the nucleic-acid-associated patterns by the innate immune system by stimulating PRRs as they can produce both RNA and DNA intermediates (reverse transcription) [78]. The ssRNA formed can be sensed by the TLR-7 and -8, thus resulting in the secretion of IFN-α by the stimulated macrophages and dendritic cells [79].

## 4. Previous Unrecorded Local Epidemics

Pakistan is a region with an insufficient focus on medical and biological sciences, especially epidemiology. The country faces many flu-based epidemics each winter season, but some are recorded with much more enthusiasm, such as the COVID-19 situation. This hypothesis is based on pure speculation with little evidence due to the lack of public records or pre-existing data in this field. Nevertheless, let us investigate the biology of how a previous epidemic may have affected the population of a region in regard to COVID-19. Having been infected with a similar upper respiratory tract flu infection that had milder effects on the host may have caused the immune system to adapt to the succeeding SARS-CoV-2 infection so that the adaptive immunity more quickly clears the infection. 

## 5. Copy Number Variation

Copy number variation (CNV) refers to the variation in the number of a specific gene repeated several times or not on the genome. Copy number variations cover about 12 percent of the genome [80]. Copy number variations of genes play a role in genetic variability and epigenetics. It has also been observed that the CNV phenomenon also plays a role in spiking up the innate and the adaptive immune systems, thus better enabling them to clear out infections (Figure 4) [81,82]. Copy number variation also seems to have a role in the expression of genes altering the normal physiological functions of organisms. For example, the expression of the alpha-amylase-1 gene is directly proportional to its dosage in an individual’s genome. This discovery was seen in a huge study involving various regions’ populations. Their genome analysis was performed, and it was seen that people of different regions had different gene dosages and copy numbers of the genome, depending on their lifestyle and food habits. Since the gene was responsible for the production of amylase enzyme, which has a crucial role in the digestion of starch, its crucial role can be observed in the life of a living organism. Therefore, the gene (AMY-1) variation in populations around the globe made scientists in the field conclude that the phenomenon may be involved in the evolutionary and adaptive change as it is related to dietary changes [82].

The CNV of the trinucleotide sequence is also seen to be a causative agent of Huntington’s disease [83], whereas in another study, it was reported that the CNV of sequence refers to the age of the onset of Huntington’s disease [84]. The evidence mentioned above is probably enough to push the idea that CNVs play a role in genetic as well as infectious diseases by regulating the immune system. The severity of COVID-19, although dependent on many factors, is highly dependent on the cytokine response of the body. A cytokine storm is mainly caused by agents such as IL-2, IL-6 TNF-α, IL-10, etc. [85]. Severe forms of cytokine response may also cause the body to go into ARDS and shock resulting in the mortality of the patient. A recent article published [62] on the overall long-term effects of COVID-19 also analyzed potential risk factors, including age, gender, acute COVID-19 severity, obesity, and pre-existing allergic diseases. Still, none of these factors could have a strong association with the long-term effects of the disease. The article also points out the lack of race/ethnicity, socioeconomic, education, and employment-based data to extrapolate the analyzed data. 

The antibody enhancement mechanism is a phenomenon in which viral entry and growth are enhanced due to the presence of specific antibodies. This phenomenon seems to play a role in the severity of infection in people who have received vaccine therapy or are reinfected with the viral disease. It is also important to mention that CNVs, such as the above-mentioned biological phenomenon, are also variable and different among people of different geographical populations [80]. Hence, we would like to represent the hypothesis that the severity of the illness of COVID-19 and its mortality may have a direct or indirect relation to the copy number variation of genes associated with the immune system and cytokine response, just to name a few. Alternatively, the CNVs of the genomic components’ other involved mechanisms may be over or underregulated, thus explaining the disparity of the COVID-19 severity and mortality among people of different geographical populations.

## 6. Potential Cell–Gene-Based Immunotherapy Human-Gene-Transfer-Based Genetic Engineering, Gene Drive Technologies

Genetic immunotherapy is a therapy that uses immune cells with modified genetics for an improved capability to fight infections or track, trace, and destroy cancer cells. Suppose the theories presented herein are correct about different ethnic population groups presenting differential genetic components that improve protection against infections and diseases. In that case, we can replicate the same process in less fortunate gene groups using the techniques and principles of genetic immunotherapy, as this strategy could provide viable immediate treatment options in case of new emerging pandemics [86].

Human-gene-therapy-based genetic research focuses on transferring a genetic component from one individual to another for therapeutic or beneficial purposes. Therefore if we can actually identify and clone genes that provide special immunity to some individuals in populations or ethnic groups, then using this technology will also allow us to give better immunological stability to other population groups [87].

In that regard, clustered regularly interspaced short palindromic repeats (CRISPR)-Cas-9-based gene editing technologies are gaining acceptability in the field of research. We can also use this technology to enhance the survival rate of genetically predisposed individuals belonging to a certain ethnic group [88].

## 7. Conclusions

The analysis and hypotheses presented in this writing may be speculative in parts. Still, it should encourage a range of new experiments and data analyses to look at the disease etiology and ecology more holistically. Such studies based on biological phenomena that we have ignored until now may, in fact, finally lead us to investigate the underlying reasons for the geographic disparity of COVID-19. Investigations into these phenomena may also help us more logically understand previous, current, or future pandemics.

## 8. Future Work Directions

The ideas presented in our paper are, of course, still theories at best even with supporting evidence from other cases. Therefore, to prove them and obtain some benefit out of them would require various research projects, and the following steps will be essential to confirm these hypotheses:The collection of strategically randomized samples of DNAs, i.e., setting up a broad spectrum of subjects from major demographic regions of the world. Important things to keep in mind while choosing subjects would be that they should be genetically diverse from each other and not just from different countries (for example, the Eastern Punjab region of Pakistan and the Western Punjab region of India have people with similar gene pools. Therefore, the results of this sample might not be so prominent). It is important to at least have subjects from major parts of Europe, North and South America, and Asia (Southeast Asian Island nations and South Asian countries included).For the virome and microbiome hypothesis, metagenomic analysis and classification and then a comparison of subjects’ microbiomes with each other’s, and finding the potentially helpful groups of bacteria, viruses and bacteriophages, may allow us to move towards unleashing the therapeutic arenas of the microbiome and virome, which include the modulation of a person’s microbiome, phage therapy, pre-, and pro-biotic therapies, microbiome transplantation, etc.HERVs will require genomic analysis and genome sequences. If we can figure out the region of HERV that is potentially boosting the immune system of a subject, we may be able to replicate it and find its molecular basis and biological causative agents involved, thus allowing us to obtain a better understanding of the disease host interaction role of ERV in our genome, all of which, in turn, may allow us to utilize the possible therapeutic benefits.To test the ENV hypothesis, an extensive genomic analysis (NextGen) would be required to identify potential genes and find their respective roles in epigenetics and immune modulation, etc., which would allow us to understand the functions of the mechanisms better.A study on the population of Pakistan and Bangladesh for hypothesis 3 can also be performed to investigate whether such a case exists or not. This will require a lot of data analysis, bioinformatics, and big data comparative analysis, but if such a case is indeed found, then we may be able to exploit the same phenomenon to potentiate relative immunity against a future pathogen with the help of the region’s previously encountered and less dangerous pathogens.Lastly, having all of this analytical big data will allow us to use AI and various algorithms to predict the effect of future SARS-CoV* strains in a certain region or a geographical location based on their biological capabilities and not just the genetic backbone, which will give the scientific and medical communities a huge edge in refocusing their limited resources.

## Figures and Tables

**Figure 1 vaccines-11-00319-f001:**
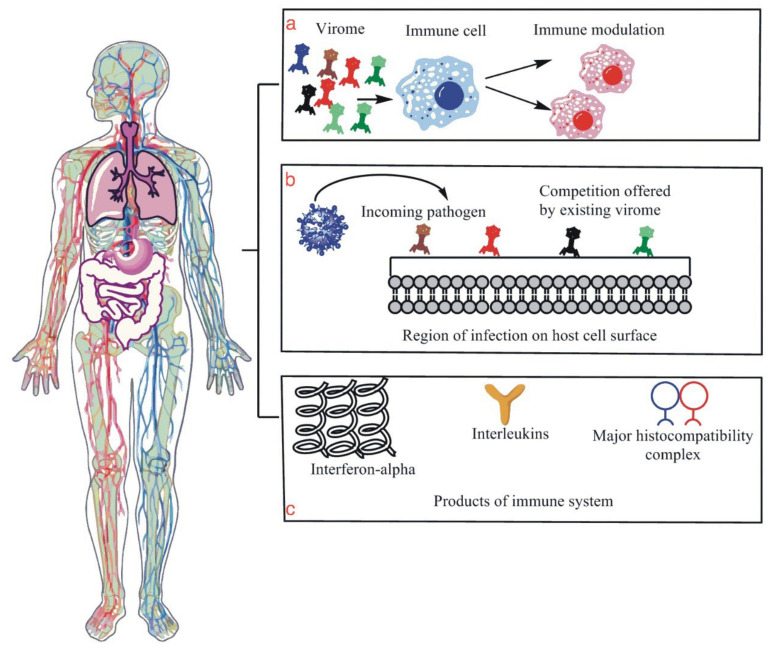
This figure is an illustration of the human microbiome, specifically the human virome and its few functions related to immunity against viral infection. (**a**) is an illustration that shows modulation or the positive reinforcement of the host immune system by virome molecules and particles that enhance the immune system’s ability to fight the incoming infection. (**b**) is a diagram illustrating the competition that is given to the incoming viral particles by the host virome, which essentially protects the host cells from infection as the virome particles act as guards of the region. (**c**) shows the production of IL, interferon, MHC molecules, all of which are essential parts of the host immune system, and their production with the help of virome demonstrates their role in the proper functioning of the host’s immune system.

**Figure 2 vaccines-11-00319-f002:**
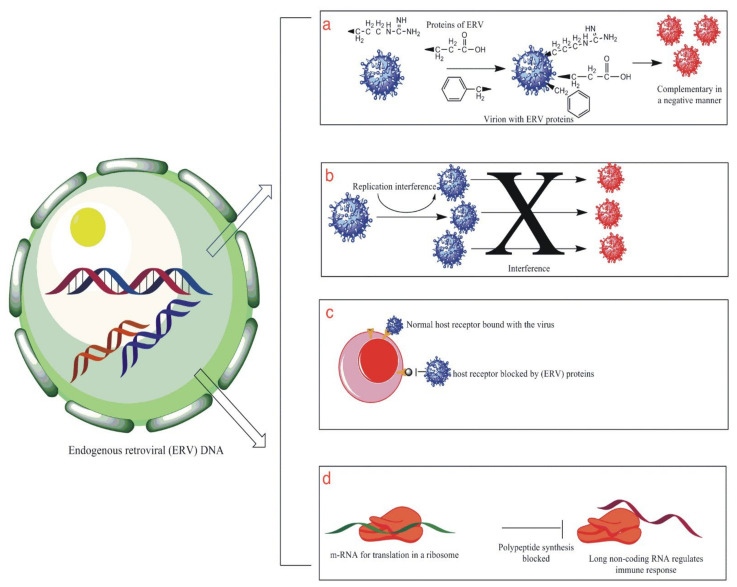
This illustrates some of the functions that HERV genome plays in protecting the human body against external viral infection. (**a**) shows the production of viral proteins by the HERV genome that are complimentary to the external virus in a negative way, thus affecting their success. (**b**) is an illustration showing the interference produced by HERV proteins that result in inhibiting or blocking the viral replication cycle. (**c**) is an illustration showing the phenomenon of receptor interference that involved the blockage of the site of entry for the foreign virus by HERV proteins, thus resulting in the prevention of viral infection of the cell. (**d**) is an illustration that shows the production of long non-coding RNA sequences that help in the modulation of the host antiviral immune system.

**Figure 3 vaccines-11-00319-f003:**
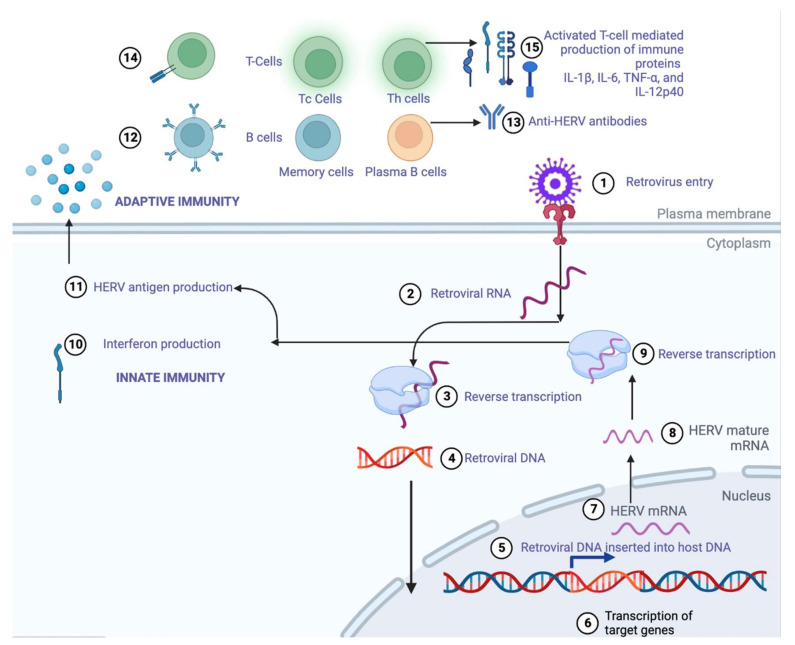
A stepwise (1–15) illustration of how HERVs can induce innate and adaptive immunity through the production of retroviral proteins that activate B and T cells to either produce anti-HERV antibodies or immune proteins IL-1β, Il-6, IL-12p40, and TNF-α to protect against newer, more evolved strains of viral pathogens.

**Figure 4 vaccines-11-00319-f004:**
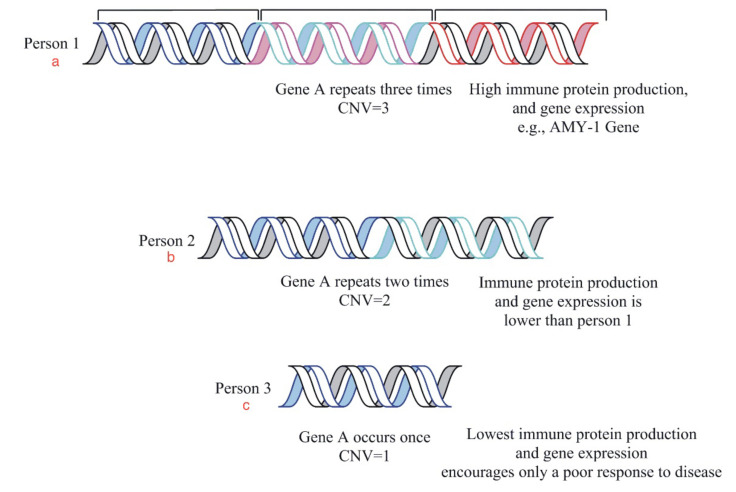
This is a diagrammatic representation of the relationship between gene dosage and gene expression in three individuals. The diagram is an example of the copy number variation phenomenon showing us the variable copy numbers of a gene in different people. The protein production is observed as (**a**) high in person 1, (**b**) moderate in 2 as compared to (**c**) low in person 3 due to the variable number of gene copies.

## Data Availability

For most data included in this article, we referred to the online, publicly available COVID-19 data-sharing resources such as WHO and Kaiser Family Foundation (KFF). As well as these resources, we also consulted peer-reviewed articles (a list is provided under the references section) that provide links and insights into the phenomena we discussed.

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
