# Peer review of "COVID-19: The Ethno-Geographic Perspective of Differential Immunity"

_vaccines, 2023, doi:10.3390/vaccines11020319_

Round 1

Reviewer 1 Report

 Authors are discussing about the heterogeneity and diversity of human microbiota which affect the variety of susceptibility and pathogenesis of COVID-19 and explain the ethno-geographic differences of COVID-19.  It seems very interesting and well described. However, the manuscript is extremely focused on the viromes and endogenous retroviruses.

There is also hygiene hypothesis that affects and explains the variety of susceptibility and pathogenesis and explain the ethno-geographic differences of COVID-19, which means that frequent or persistent infection, viral or bacterial, and the presence of various parasites in the body may be involved in the different clinical course of COVID-19 from the ethno-geographic point of view, just like allergic diseases.

I think this perspective is missing in this manuscript.  This hygiene hypothesis is also related with the differences and heterogeneity of viromes and endogenous retroviruses in some way.

I recommend to overview or comment on this whole perspective at least in the introduction, and proceed to focus on the viromes, endogenous retroviruses, non-coding RNAs and so on, because this manuscript is a review article, which requires board perspective for the readers.

Author Response

Authors are discussing about the heterogeneity and diversity of human microbiota which affect the variety of susceptibility and pathogenesis of COVID-19 and explain the ethno-geographic differences of COVID-19.  It seems very interesting and well described. However, the manuscript is extremely focused on the viromes and endogenous retroviruses.

There is also hygiene hypothesis that affects and explains the variety of susceptibility and pathogenesis and explain the ethno-geographic differences of COVID-19, which means that frequent or persistent infection, viral or bacterial, and the presence of various parasites in the body may be involved in the different clinical course of COVID-19 from the ethno-geographic point of view, just like allergic diseases.

I think this perspective is missing in this manuscript.  This hygiene hypothesis is also related with the differences and heterogeneity of viromes and endogenous retroviruses in some way.

I recommend to overview or comment on this whole perspective at least in the introduction, and proceed to focus on the viromes, endogenous retroviruses, non-coding RNAs and so on, because this manuscript is a review article, which requires board perspective for the readers.

  • We have added the hygiene hypothesis to potentiate more clarity and focus to the various hypotheses we have proposed in this review article. This is a very valid point and we have added a sub-heading based on this point (sub-heading 7 in the article).

Reviewer 2 Report

The author attempts to review genetic, epigenetic, and molecular immune resistance mechanisms in correlation with the pathophysiology of SARS-CoV-2 and the varied ethnicity-based immunological responses against it. Authors have discussed the overall landscape of human microbiota, endogenous retroviral genes spliced into the human genome, copy number variation, and how they could modulate the innate and adaptive immune systems, which put a particular ethnic genetic architecture at a higher risk of SARS-CoV-2 infection than others. However, in my opinion, the paper has some shortcomings regarding some data analysis and the text. Below, I have provided numerous remarks on the text, as it is often vague and long-winded.

  1. Title should be replaced from COVID-19: the ethno-geographic perspective of differential im- munity. to “COVID-19: The Ethno-Geographic Perspective of Differential Immunity
  2. Authors should include more data and investigations or add one table of studies carried out to dissect the role of the human microbiome in understanding SARS-CoV-2 infection and disease progression.
  3. Authors should explain the upregulated expression of multiple human endogenous retrovirus families in senescence induced HBECs (human bronchial epithelial cells) in comparison to that in noninduced HBECs, a fact that could possibly explain the differences in disease severity among age groups.
  4. Authors should include the mechanism by which ERV acts as a genetic memory and triggers the innate immune system's sensing.
  5. The authors should explain with a diagrammatic representation the modulation of the immune system by ERV-based proteins.
  6. Authors should include Innate Lymphoid Cells: Diversity, Plasticity, and Unique Functions in Immunity.
  7. Please check once again for any flaws throughout the text, standard abbreviations, and references.

I have corrected typos, grammar, and spelling errors in the abstract; authors are requested to please check throughout the text.

Coronavirus disease 2019 (COVID-19), the emissary behind the worst global pandemic of the 21st century, is primarily a respiratory disease-causing virus called SARS-CoV-2, which is responsible for millions of new cases (incidents) and deaths (mortalities) worldwide. Many factors, such as the quality of primary medical health care facilities or enabling economies, have contributed to the disparities in morbidity and mortality experienced by nations and ethnicities in response to SARS-CoV-2. Nevertheless, the most important variable, i.e., the subsequent ability of individuals to be immunologically sensitive or resistant to the infection, was not properly discussed before. Therefore, an astounding issue arose when some developed countries experienced higher morbidity and mortality compared with their relatively underdeveloped counterparts, despite having excellent medical health facilities. Hence, this investigative review attempts to analyse the issue from the angle of previously undiscussed genetic, epigenetic, and molecular immune resistance mechanisms in correlation with the pathophysiology of SARS-CoV-2 and varied ethnicity-based immunological responses against it. The biological factors discussed here include the overall landscape of human microbiota, endogenous retroviral genes spliced into the human genome, copy number variation, and how they could modulate the innate and adaptive immune systems, which put a particular ethnic genetic architecture at a higher risk of SARS-CoV-2 infection than others. Considering an array of these factors in their entirety may help explain the geographic disparity of disease incidence, severity, and subsequent mortality associated with the disease while at the same time encouraging scientists to design new experimental approaches to investigation.

Author Response

Reviewer 2:

  1. Title should be replaced from COVID-19: the ethno-geographic perspective of differential im- munity. to “COVID-19: The Ethno-Geographic Perspective of Differential Immunity”
  • The first letter of each word has been capitalized as suggested by the reviewer.

2. Authors should include more data and investigations or add one table of studies carried out to dissect the role of the human microbiome in understanding SARS-CoV-2 infection and disease progression.

  • We have included almost 10 different studies showing a direct correlation between different components of the human microbiota and COVID-19 disease immunity and progression [please review new references 25-35].

3. Authors should explain the upregulated expression of multiple human endogenous retrovirus families in senescence induced HBECs (human bronchial epithelial cells) in comparison to that in noninduced HBECs, a fact that could possibly explain the differences in disease severity among age groups.

  • Upregulated expression of HERVs in senescence induced HBECs was added. However, more detail was intentionally not included due to the unrelatedness of the suggestion to our manuscript as we are focusing more on the geographical and ethnic disparity instead of senescence.

4. Authors should include the mechanism by which ERV acts as a genetic memory and triggers the innate immune system's sensing.

  • We have added a section regarding genetic memory and innate immune system triggering by HERV.

5. The authors should explain with a diagrammatic representation the modulation of the immune system by ERV-based proteins.

  • Added a new illustration (figure 3) to show how ERV-based proteins can modulate the immune system.

6. Authors should include Innate Lymphoid Cells: Diversity, Plasticity, and Unique Functions in Immunity.

  • We have added a brief overview on ILCs which is related to our manuscript and provided a much clearer supporting explanation of the hypotheses being put forward by this investigative review article.

7. Please check once again for any flaws throughout the text, standard abbreviations, and references.

  • All authors checked the grammar and sentence structure again and removed a few errors.

Round 2

Reviewer 1 Report

I think that the manuscript was revised well in response to the reviewer's comment.

I am glad to agree to accept the revised manuscript.

Reviewer 2 Report

Accept in present form